# Differences between returns to education in Urban and rural China and its evolution from 1989 to 2019

Xing Gao[1], Maishou Li[1,2]*

**1** School of Economics, Institute of Economics, Henan University, Kaifeng, Henan, China, **2** Institute of Rural Revitalization, Henan University, Kaifeng, Henan, China

* lms@henu.edu.cn

**Data Availability Statement:** All relevant data are within the manuscript and its Supporting information files.

## Abstract

The income gap between urban and rural residents has long been a predicament for China. The differences between returns to education in urban and rural China are one of the important factors affecting the income gap. Using a combination of data from CHNS, CHIP, CGSS, CFPS, CHFS, and CSS, the differences in returns to education and its evolution in China from 1989 to 2019 were estimated. Results show that returns to education in urban China have been consistently higher than that in rural China. Returns to education in urban China show a trend of progressive increase, then a rapid rise, before turning into a slow decline and gradually leveling off; returns to education in rural China exhibit a slowly increasing trend before gradually leveling off; the differences between returns to education in urban and rural China show an evolution of first growing larger, then smaller, before gradually leveling off. The spouse's education was considered the instrumental variable of individuals' education. The robustness test was done with an estimation through a two-stage least squares (2SLS) method. Results indicate that the empirical conclusion has good robustness. The evolution of returns to education in China was explained in terms of the marketization of labor forces, the relative supply and demand of labor forces, the reform of the household registration system, and the evolution of the quality of education.

## Introduction

Since the implementation of the reform and opening up policies in 1978, China has had a rapidly growing economy, which has greatly improved the living standards of its citizens. In this period, the distribution of educational opportunities in China has undergone great changes, and education at all levels has achieved comprehensive development. However, China's rapid economic and educational development has resulted in inequality in income distribution. The income gap between urban and rural residents is the primary cause of the country's overall income disparity [1]. With China's achievements in poverty alleviation and the construction of an overall moderately prosperous society, the relationship between urban and rural areas in China has been gradually developing toward a symbiotic relationship, and the system and mechanism for the integrated development of urban and rural areas have basically taken

**Funding:** The author(s) received no specific funding for this work.

**Competing interests:** The authors have declared that no competing interests exist.

**Abbreviations:** CFPS, (China Family Panel Studies): http://www.isss.pku.edu.cn/cfps/; CGSS, (Chinese General Social Survey): http://cgss.ruc. edu.cn/; CHFS, (China Household Finance Survey): https://chfs.swufe.edu.cn/sjzx/sjsq.htm; CHIP, (Chinese Household Income Project Survey): http://ciid.bnu.edu.cn/; CHNS, (China Health and Nutrition Survey): https://www.cpc.unc.edu/ projects/china; CSS, (Chinese Social Survey): http://csqr.cass.cn/index.jsp.

shape. Rural development is accelerating, the income of farmers is rising rapidly, and the income gap between urban and rural residents is narrowing year by year. However, for historical reasons, China's urban-rural dual economic structure has not fundamentally changed, and the income gap between urban and rural areas remains extremely wide. In 2019, the per capita disposable income of urban residents was 2.64 times that of rural residents (Data Source: 2020 China Statistical Yearbook). In the process of the marketization of labor forces in China, education has become an important factor affecting income levels. Correspondingly, the gap between educational levels in urban and rural areas is also one of the major factors contributing to the disparity in income levels between urban and rural areas [2]. The gap between educational levels in urban and rural China accounted for more than one-third of the factors affecting the income gap between urban and rural China [3]. Education in China is always evolving, but the educational level in rural areas has always been lower than that in urban areas. The lower educational level of rural residents contributes to the low overall educational level in China [4]. Education is the most important form of investment in human capital, and investment in education has significant economic value for individuals. Returns to education reflect the increase in labor income brought about by higher educational attainment. The higher the returns to education, the stronger the incentive to invest in human capital. In the past few decades, the educational level of the Chinese has risen rapidly, which is closely related to the increase in the returns to education. As China has now entered a new development stage, the high-quality economic development in China must increase returns to education and formulate effective incentives for investment in human capital, thereby enhancing the quality of the demographic dividend. Given the regional nature of the labor market in China, differences exist between returns to education in urban and rural China, which not only determine the mobility mode of labor forces with different educational levels but will also continuously affect the pattern of income distribution and social stratification. Thus, studying the relationship between educational level and income, especially the evolution of the differences between urban and rural areas in China, has important and practical significance for accelerating the accumulation of human capital, improving the efficiency of educational resource allocation, rationally formulating policies for educational development and public finance, and optimizing the decision making of individuals' education.

The research on the returns to education is based on human capital theory. Since Mincer proposed the wage determination equation in the 1970s, the Mincerian rate of return has become the main method for measuring the economic value of education. By estimating the Mincer equation, scholars obtained the returns of educational elements. Since the 1990s, scholars have begun to pay attention to the estimate of the returns to education in China, resulting in numerous studies that have been conducted using various models and methods [5, 6]. When estimating the returns to education, it is assumed that all individuals or groups have the same returns to education, and the estimated value of the education coefficient is a constant for all the individuals or groups. However, in reality, different individuals or groups are heterogeneous. Therefore, an increasing number of researchers are focusing on analyzing the heterogeneity of returns to education. The differences between returns to education in urban and rural China have also become a focus of scholars because of the country's unique urban-rural dual economic structure. In the early 1990s, returns to education in urban and rural China were extremely low and had no significant difference [7, 8]. At the beginning of the 21st century, the returns to education in China had increased but remained relatively low; the returns to education in urban China were roughly 8%, whereas the returns to education in rural China were only nearly 4% [9]. After Zhang controlled for the related personal characteristic factors, the returns to education in urban China were 3.09% higher than that in rural China [10]. The empirical results of Meng and Xiong showed that the difference between returns to education

in urban and rural China was roughly 2% [11]. In addition to the significant difference between returns to education in urban and rural China, the gap between the returns to education in urban and rural China increased with the surge in income quantile [12]. The returns to education in urban China were also significantly higher than that in rural China [13]. Liu et al. found that even after addressing the mismeasurements of wage rate and experience as well as self-selection, the return to rural schooling in China remained low [14]. Generally, the research on the differences between returns to education investment in urban and rural China found that the returns to education in urban China are higher than that in rural China. However, the above-mentioned studies used the cross-sectional data of a specific year to statically analyze the differences between returns to education in urban and rural China. Meanwhile, the evolution of returns to education in China has gradually attracted extensive attention from scholars. The returns to education in urban China increased by nearly three times from the early 1990s to the late 1990s [15]. The empirical study by Huang showed that the returns to education in China increased from an insignificant rate in 1989 to 11% in 2000 [16]. Meanwhile, Fan found a significant increase in the returns to education in China from 2003 to 2008 [17]. However, since the middle and late 21st century, the average returns to education in urban China have shown a steady upward trend rather than a rapid upward trend [18]. Concurrently, from 2007 to 2013, the returns to education in rural China only slightly increased [19].

Most existing studies used cross-sectional data from a specific year for research or dynamically analyzed the returns to education in urban China within a relatively short time span, failing to account for the long-term evolution of returns to education in urban and rural China. In addition, China has a relatively complex urban-rural dual structure, and only a few studies have been conducted on the long-term evolution of returns to education in rural areas, as well as the difference between returns to education in urban and rural areas. Therefore, this paper analyzes the differences between returns to education in urban and rural areas in China and their evolution.

## Model, data, and variables

### Empirical model

The Mincer income equation is widely used by researchers in studies on returns to education. The following empirical model is constructed on the basis of the Mincer income equation.

$$\text{LnIncome}_i = \alpha_0 + \alpha_1 \text{Edu}_i + \alpha_2 \exp_i + \alpha_3 \exp_i^2 + X_i' \beta + \varepsilon_i \tag{1}$$

Where Income represents the income of an individual, and LnIncome is the logarithm of income; Edu represents the educational level of the individual, measured by the years of education; exp represents the work experience of the individual, and $\exp^2$ is the squared term of the work experience of the individual; X represents other control variables, and $\varepsilon$ is the random disturbance term. The parameter estimator $\alpha_1$ is the return to education.

In the research on returns to education, endogeneity must be discussed due to the missing unobservable factor. People with higher abilities often acquire higher incomes and are more inclined to receive more education. Thus, ability is related to income and education. However, ability is an intangible factor that we cannot directly observe and thus is classified as a disturbance in the model. In this way, the disturbance is related to education, leading to endogeneity problems. A commonly used solution is the instrumental variable (IV) method. However, this method requires meeting two conditions. The first condition is a correlation; that is, instrumental variables should be related to education. The second condition is exogeneity; that is, the instrumental variables are not related to the disturbance. Whether the exogeneity is conformed to can be determined by discussing whether the "exclusive constraints" are satisfied;

that is, the endogenous explanatory variable is the only channel for instrumental variables to affect outcome variables. This paper focuses on the long-term evolution of the returns to education rather than emphasizing the level of the returns to education at a particular period. If the correlation coefficient between the unobservable ability factor and the educational level and income level has a fixed or approximately fixed time trend, the unobservable ability factor does not affect the analysis of the changing trend of the returns to education [20]. However, as a robustness test, the instrumental variable (IV) method was still considered to alleviate endogeneity problems. In previous studies, parents' education [21, 22] and the distance from home to the university [23] are often used as instrumental variables for the years of an individual's education. However, some scholars have questioned the rationality of these instrumental variables, given that although they are related to educational level, proving that they are independent of the disturbance term is difficult [24]. A classic instrumental variable of education is the quarter of the year in which an individual is born [25, 26], but it may not be applicable to China's situation [27]. Furthermore, the dynamic analysis in this paper covers a relatively long time span; thus, searching for the same instrumental variable for the estimation of returns to education for each year is necessary. On the basis of the operational availability and existing research [28, 29], the spouse's education is used as the instrumental variable for individuals' education in this paper. People get married because they share similar interests or behavioral characteristics [30]. Li discovered that the degree of matches between a couple's educational levels began to rise rapidly in the 1980s and has remained high since then [31]. However, the proportion of spouses with different education levels is decreasing. More people have the same or similar educational level as their spouses. Therefore, the educational levels of a couple generally do not differ significantly and are closely related. Concurrently, the educational level of the spouse cannot directly affect the income level of the individual; only the individual's educational level can affect their income, which satisfies the exclusive constraint condition. Guo et al. concluded that a spouse's education is a strong instrumental variable [32]. Therefore, the educational level of the spouse can be considered a good instrumental variable. When the educational level of the spouse is used as the instrumental variable, the sample is restricted to the married sample. Using the spouse's years of education as the instrumental variable is controversial, which means that the educational level of the spouse may not be the best choice for the instrumental variable. However, due to the long time span of the data used in this chapter, applying instrumental variables on the basis of policy implementation to the data of each year is not feasible. Therefore, the educational level of the spouse is relatively better than other instrumental variables.

In the following part, the IV method is used for the robustness test. Thus, the IV model is constructed.

$$\text{Edu}_i = \alpha_0 + \alpha_1 Z_i + \alpha_2 \exp_i + \alpha_3 \exp_i^2 + X_i'\beta + \varepsilon_i \tag{2}$$

where Z represents instrumental variables of education.

## Data sources

For data continuity and long time span, several micro databases in China are combined in this paper, and the same model and estimation method are employed for analysis. Although only a few studies combined several micro databases, many of them were successful. For example, Li et al. combined data from CHNS, CGSS, CULS, and CHIP from 2000 to 2009 to analyze the impact of financial development on entrepreneurship [33].

The microdata used in this paper includes data from CHNS, CHIP, CGSS, CFPS, CHFS, and CSS from 1989 to 2019, of which data used are from CHNS for 1989, 1991, 1993, 1997,

2000, 2004, and 2009; from CHIP for 1995, and 2002; from CGSS for 2005, 2006, 2008, 2010, 2011, 2012, 2013, and 2015; from CFPS for 2014, 2016, and 2018; from CHFS for 2017; from CSS for 2019. CHNS was established in collaboration with the Carolina Population Center at the University of North Carolina at Chapel Hill and the National Institute for Nutrition and Health at the Chinese Center for Disease Control and Prevention, and is a longitudinal data; CHIP is a survey conducted by Beijing Normal University and domestic and foreign experts, and is a cross-sectional data; CGSS is initiated and implemented by Renmin University of China, and is a cross-sectional data; CFPS is initiated and implemented by Peking University of China, and is a longitudinal data; CHFS is initiated and implemented by Southwestern University of Finance and Economics of China, and is a longitudinal data; CSS was initiated and implemented by the Chinese Academy of Social Sciences, and is a cross-sectional data. The sampling of each micro data is nationally representative, hence, they are reliable data sources. This paper focuses on the difference between returns on education in urban and rural China. Therefore, the samples who are still studying in school, self-employed, self-employed in the industrial and commercial sectors, entrepreneurs, working in family-owned businesses, and household workers, and samples with missing variables are removed, whereas the samples whose ages are between 18 and 60 are retained.

### Description of variables

The dependent variable is the individual's income. The indicators of income used in previous studies mainly include annual income, monthly income, and hourly income. Chinese people are more concerned about their total annual income [34]. In addition, most of the six micro databases used in this paper included the annual income of individuals as an indicator. Therefore, the dependent variable used in this paper is the annual income (including bonuses and subsidies). To eliminate the influence of inflation, the annual income is adjusted on the basis of the Consumer Price Index published by each province. CPI data comes from the China Statistical Yearbook. Divide nominal annual income by CPI to obtain the real annual income adjusted for inflation.

The core independent variable is the years of an individual's education (Edu). Some of the six micro databases straightforwardly state the years of education, which can be used directly in this study, such as CHNS and CFPS, whereas some of them only state the educational level, such as CGSS and CSS, which must be converted into corresponding years of education. For an individual's work experience (exp), "age of an individual–years of education–6.," which is typically practiced in most literature, is used in this study. For other control variables, as several micro databases are used and each has different indicators, gender is selected to be the control variable. In the regression, the provinces' fixed effect is controlled. The household registration system is the common basis for the division of personnel in urban and rural areas, and it can help identify the characteristics and problems brought by the system. Therefore, in this paper, the division of samples is conducted on the basis of their registered residence. Table 1 shows the descriptive statistics of each variable.

## Empirical results and analysis

### Empirical results

Table 2 reports the OLS estimation results of returns to education in urban and rural China and their differences from 1989 to 2019. From a static perspective, at any time point, returns to education in urban China are significantly higher than that in rural China, which is consistent with the conclusion of most studies on the differences between returns to education in urban and rural China. Based on the regression results of urban samples, returns to education in

**Table 1. Descriptive statistics: Mean.**

| | Variables | 1989 | 1991 | 1993 | 1995 | 1997 | 2000 | 2002 | 2004 | 2005 | 2006 | 2008 |
|---|---|---|---|---|---|---|---|---|---|---|---|---|
| Urban | Income | 803.46 | 867.60 | 1253.28 | 1952.56 | 1516.04 | 2198.80 | 2831.54 | 3164.97 | 3061.26 | 3637.95 | 4304.26 |
| | Edu | 9.29 | 9.58 | 9.87 | 11.63 | 10.45 | 11.25 | 12.04 | 11.52 | 10.99 | 11.71 | 11.94 |
| | exp | 19.10 | 19.01 | 19.89 | 22.87 | 20.52 | 20.61 | 24.24 | 22.28 | 24.80 | 23.32 | 22.43 |
| | gender | 0.55 | 0.54 | 0.56 | 0.51 | 0.58 | 0.58 | 0.51 | 0.58 | 0.47 | 0.48 | 0.53 |
| | Observations | 1492 | 1375 | 1737 | 12045 | 1547 | 1640 | 12304 | 1243 | 3438 | 2721 | 1800 |
| Rural | Income | 785.39 | 779.41 | 875.52 | 1588.88 | 1335.08 | 1777.59 | 1097.22 | 2113.42 | 1223.28 | 2491.27 | 1593.11 |
| | Edu | 8.51 | 8.87 | 8.04 | 9.22 | 8.34 | 8.81 | 8.99 | 9.11 | 7.22 | 8.84 | 7.63 |
| | exp | 17.86 | 18.13 | 17.39 | 19.01 | 18.12 | 17.97 | 20.84 | 21.19 | 26.56 | 19.55 | 25.56 |
| | gender | 0.62 | 0.60 | 0.63 | 0.65 | 0.60 | 0.64 | 0.71 | 0.62 | 0.46 | 0.54 | 0.50 |
| | Observations | 1388 | 1269 | 622 | 1252 | 612 | 691 | 9688 | 401 | 3025 | 1076 | 1439 |
| | Variables | 2009 | 2010 | 2011 | 2012 | 2013 | 2014 | 2015 | 2016 | 2017 | 2018 | 2019 |
| Urban | Income | 5615.63 | 5645.89 | 5801.55 | 7158.49 | 8059.69 | 6150.32 | 9106.87 | 6966.22 | 9582.79 | 8850.12 | 8978.26 |
| | Edu | 11.92 | 13.03 | 12.89 | 13.02 | 13.32 | 12.29 | 13.06 | 12.54 | 13.42 | 13.01 | 13.65 |
| | exp | 23.99 | 20.92 | 21.81 | 21.66 | 20.97 | 20.86 | 22.78 | 17.42 | 22.05 | 20.97 | 21.57 |
| | gender | 0.57 | 0.56 | 0.57 | 0.58 | 0.61 | 0.57 | 0.51 | 0.53 | 0.56 | 0.56 | 0.49 |
| | Observations | 1251 | 1895 | 880 | 2056 | 1810 | 3322 | 1475 | 1314 | 13003 | 3197 | 1180 |
| Rural | Income | 3778.39 | 2228.85 | 2663.32 | 3414.82 | 4343.32 | 4947.67 | 5387.01 | 5109.41 | 6363.93 | 6567.45 | 5704.51 |
| | Edu | 8.99 | 7.77 | 8.07 | 8.31 | 8.54 | 8.42 | 8.86 | 9.09 | 10.12 | 9.56 | 9.98 |
| | exp | 22.70 | 26.59 | 26.52 | 26.68 | 26.19 | 19.92 | 26.47 | 17.78 | 22.16 | 20.77 | 23.27 |
| | gender | 0.63 | 0.54 | 0.50 | 0.55 | 0.56 | 0.62 | 0.53 | 0.58 | 0.60 | 0.60 | 0.50 |
| | Observations | 633 | 2604 | 1454 | 2733 | 2721 | 5783 | 2636 | 3104 | 15757 | 5797 | 1762 |

**Data Source**: CHNS, CHIP, CGSS, CFPS, CHFS, and CSS

**Table 2. Differences in returns to education between urban and rural China (OLS estimates).**

| Year | Urban | Rural | Difference | Year | Urban | Rural | Difference |
|---|---|---|---|---|---|---|---|
| 1989 | 0.0296*** (0.0034) | 0.0061 (0.0051) | 0.0235*** | 2009 | 0.0991*** (0.0063) | 0.0315*** (0.0102) | 0.0676*** |
| 1991 | 0.0252*** (0.0031) | 0.0203*** (0.0053) | 0.0049 | 2010 | 0.1151*** (0.0060) | 0.0383*** (0.0064) | 0.0768*** |
| 1993 | 0.0290*** (0.0046) | 0.0008 (0.0086) | 0.0282*** | 2011 | 0.1097*** (0.0100) | 0.0381*** (0.0093) | 0.0716*** |
| 1995 | 0.0472*** (0.0013) | 0.0183** (0.0074) | 0.0289*** | 2012 | 0.1006*** (0.0052) | 0.0466*** (0.0057) | 0.0540*** |
| 1997 | 0.0316*** (0.0039) | 0.0021 (0.0090) | 0.0295*** | 2013 | 0.0961*** (0.0069) | 0.0392*** (0.0061) | 0.0569*** |
| 2000 | 0.0650*** (0.0058) | 0.0171* (0.0088) | 0.0486*** | 2014 | 0.0809*** (0.0049) | 0.0324*** (0.0039) | 0.0485*** |
| 2002 | 0.1106*** (0.0026) | 0.0586*** (0.0055) | 0.0519*** | 2015 | 0.1053*** (0.0068) | 0.0594*** (0.0060) | 0.0459*** |
| 2004 | 0.0895*** (0.0066) | 0.0345*** (0.0131) | 0.0550*** | 2016 | 0.0858*** (0.0096) | 0.0472*** (0.0055) | 0.0386*** |
| 2005 | 0.1055*** (0.0048) | 0.0500*** (0.0058) | 0.0555*** | 2017 | 0.1126*** (0.0027) | 0.0763*** (0.0023) | 0.0363*** |
| 2006 | 0.0795*** (0.0050) | 0.0232*** (0.0088) | 0.0563*** | 2018 | 0.0852*** (0.0045) | 0.0454*** (0.0037) | 0.0398*** |
| 2008 | 0.1097*** (0.0063) | 0.0571*** (0.0089) | 0.0526*** | 2019 | 0.0984*** (0.0076) | 0.0495*** (0.0094) | 0.0490*** |

*Note*: Given the large number of years used and limited paper length, only the estimated results of education are reported, and individual work experience, work experience square terms, gender, and province fixed effects are controlled in the regression; numbers in parentheses denote robust standard error;

* significant at 10%,

** significant at 5%,

*** significant at 1%.

Complete regression results are shown in Tables 4 and 5 of the S1 Appendix.

**Data Source**: CHNS, CHIP, CGSS, CFPS, CHFS, and CSS

urban China for all years are statistically significant at 1%, indicating that education plays an important and positive role in increasing the income of urban residents. After 30 years of economic reform and development, the returns to education in urban China have also undergone tremendous changes. In 1989, returns to education in urban China were only 2.96%, demonstrating that the income of urban residents would only increase by 2.96% when years of education increased by one year. From 1991 to 1993, returns to education in urban China were also less than 3%. From the late 1980s to the early 1990s, even for urban residents, education did not play a significant role in increasing income. Since 1995, returns to education in urban China have gradually increased. In 2008, returns to education in urban China increased to 10.97%. In 2019, returns to education in urban China reached 9.84%. Compared to the late 1980s and early 1990s, returns to education in urban China today have nearly quadrupled.

Based on the estimated results of rural samples, although the coefficient of education obtained from rural samples in 1989 and 1993 were positive, they were not statistically significant, indicating that education minimally affected the income and the improvement of marginal productivity of rural residents from the 1980s to early 1990s. Since the late 1990s, education has been influencing the income of rural residents, and the improvement in educational levels has contributed to the income increase. Before 2000, returns to education in rural China were less than 2%. However, in 2019, returns to education in rural China have increased to 4.95%. That is, when the years of education of rural residents increase by one year, their income will increase significantly by 4.95%, indicating that returns to education in rural China have increased substantially in the last 30 years.

To intuitively understand the evolution of returns to education in urban and rural China and their differences, the estimation results are shown in Figs 1–3. Fig 1 presents the evolution of returns to education in urban China. The dynamic trend of returns to education in urban China fluctuates significantly, but the evolution of returns to education in urban China can be

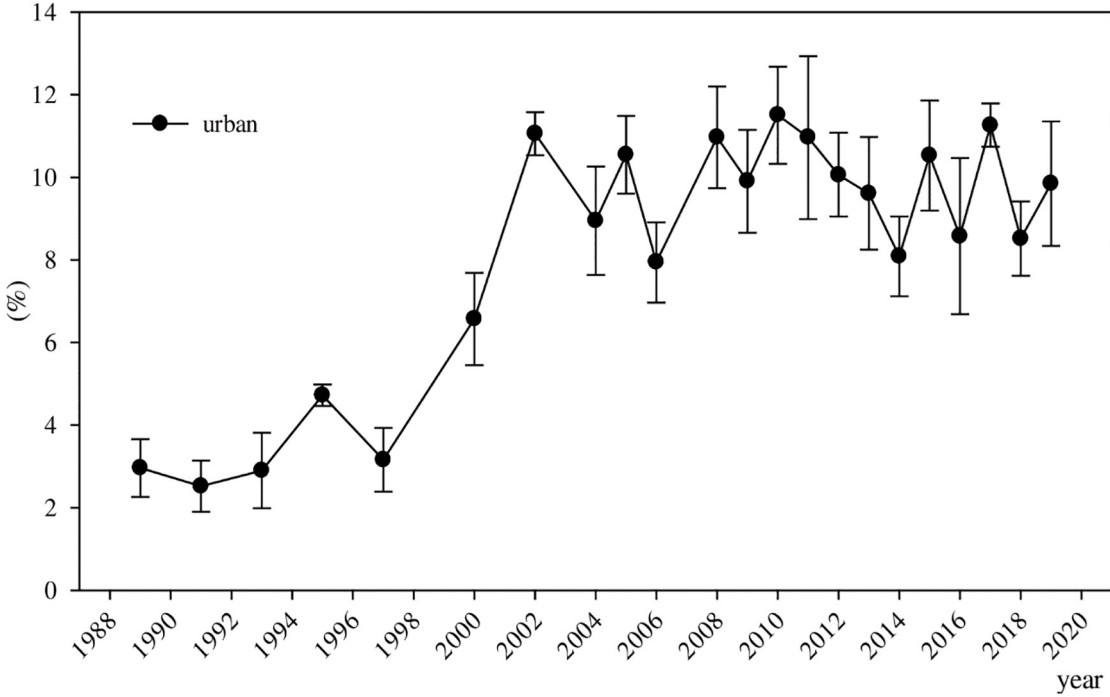

**Fig 1. Evolution of returns to education in urban China.** *Data Source*: CHNS, CHIP, CGSS, CFPS, CHFS, and CSS.

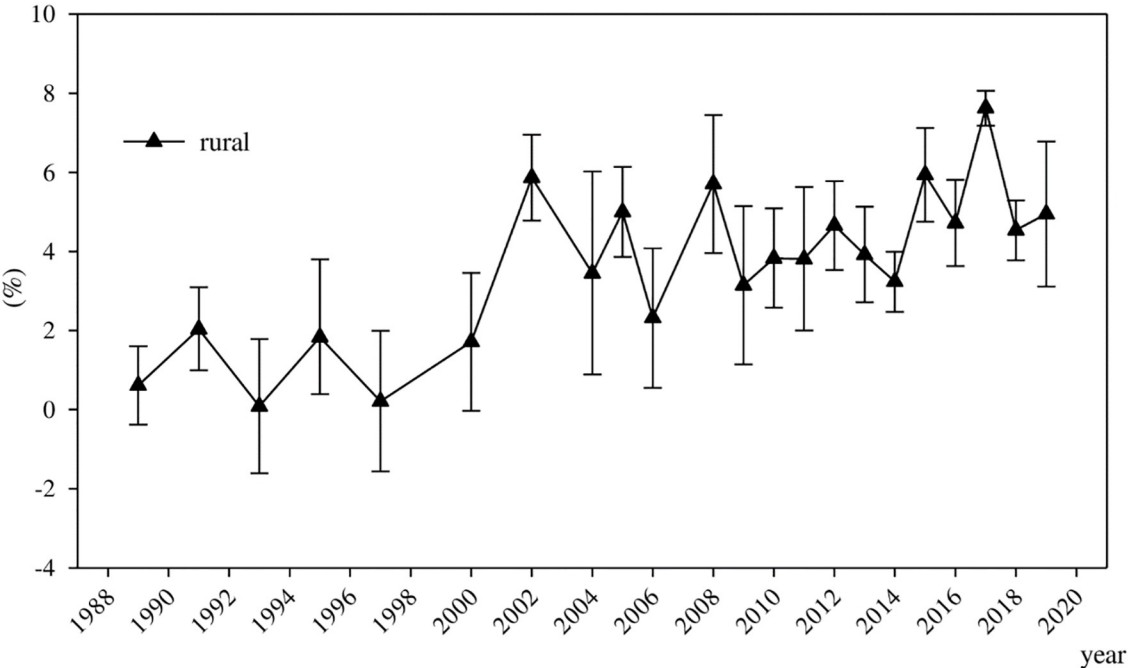

**Fig 2. Evolution of returns to education in rural China.** *Data Source*: CHNS, CHIP, CGSS, CFPS, CHFS, and CSS.

divided into three stages. The first stage is before 1993, when returns to education in urban China were less than 3%, which was basically extremely low. The second stage is after 1993, when returns to education in urban China began to grow significantly. From 1993 to 2002, returns to education in urban China increased rapidly, from 2.90% to 11.06%. After 2002,

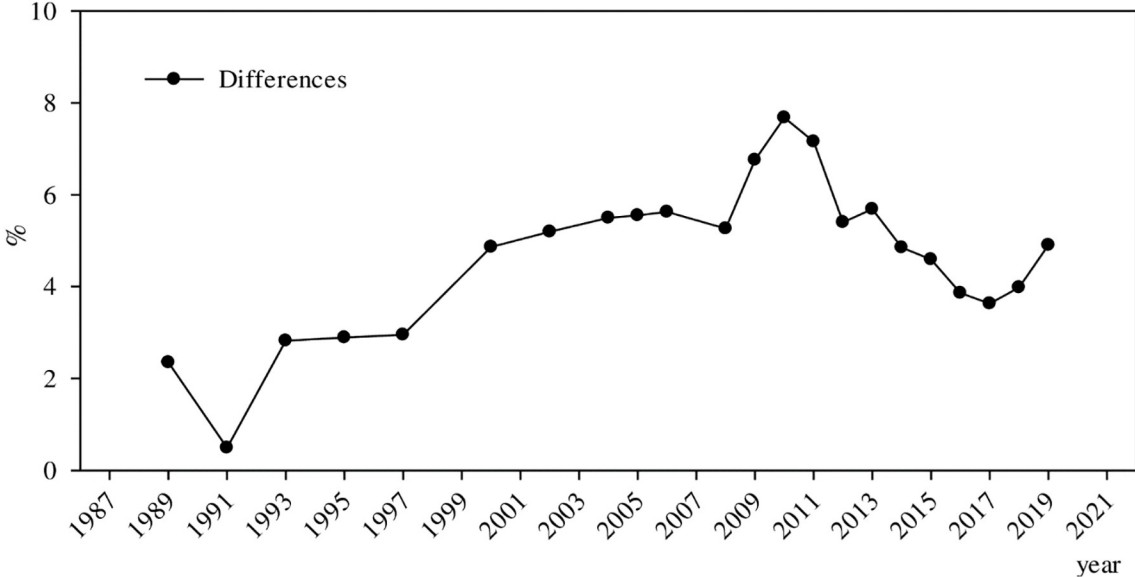

**Fig 3. Evolution of the differences in returns to education between urban and rural China.** *Data Source*: CHNS, CHIP, CGSS, CFPS, CHFS, and CSS.

returns to education in urban China no longer continued the previous trend of rapid growth and remained in a stable fluctuation. By 2010, returns to education in urban China reached a maximum of 11.51%. In the third stage, after 2007, returns to education in urban China began to decrease or even stagnate. From 2010 to 2014, returns to education in urban China slowly decreased from 11.51% to 8.09%. Since 2015, returns to education in urban China have fluctuated and slightly increased, reaching 9.85% in 2019, but remaining lower than the 11.51% in 2010. Although returns to education in urban China have a tendency to increase, the recent changes from 2013 to 2019 reveal a minimal increase. Therefore, in the past 30 years, returns to education in urban China have shown an evolution of first increasing slowly, then rapidly rising, before turning into a slow decline and gradually leveling off.

Fig 2 shows the evolution of returns to education in rural China. Based on the dynamic trend of returns to education in rural China, although returns to education in rural China fluctuate, the fluctuation is relatively small when compared to that in urban areas. Overall, returns to education in rural China show a slow growth trend. Returns to education in rural China increased from less than 1% in 1989 to 4.95% in 2019. However, compared with that in the stage of increasing returns to education in urban China, returns to education in rural China increased at a slower rate. In recent years, returns to education in rural China have been fluctuating steadily. Therefore, in the past 30 years, returns to education in rural China have shown an evolution of slowly increasing before gradually leveling off.

Fig 3 exhibits the evolution of differences between returns to education in urban and rural China. Overall, returns to education in urban and rural China have shown a trend of first increasing and then decreasing. From 1989 to 1993, the difference between returns to education in urban and rural China briefly narrowed before growing larger, but at this time, the difference is insignificant. This indicates that from the 1980s to the early 1990s, education had a minimal influence on the income of an individual in urban and rural China. After 1993, the difference between returns to education in urban and rural China gradually widened and increased from less than 2% to 7.68%, reaching its peak in 2010. During this period, compared with rural residents, education played a greater role in increasing the income of urban residents. In 2011, the difference between returns to education in urban and rural China decreased, but it did not change significantly from 2012 to 2013, remaining at around 5%. Since 2014, the difference between returns to education in urban and rural China has begun to narrow considerably. Since 2017, although the difference between returns to education in urban and rural China has slightly increased, it has remained between 4% and 5%. Therefore, overall, the difference between returns to education in urban and rural China shows an evolution of first increasing, then decreasing, before gradually leveling off.

## Robustness test

Due to the endogeneity problem, the educational level of the spouse was selected as the instrumental variable for the robustness test. The 2SLS method was used for re-estimation and the differences between returns to education in urban and rural China are shown in Fig 4.

According to Fig 3, the difference between returns to education in urban and rural China first grew larger, then smaller, before gradually leveling off in recent years. Compared with Fig 2, the main difference is in the inflection point of change. In Fig 2, the inflection point surfaced in 2010, whereas in Fig 3, the inflection point appeared in 2009. However, the overall evolution trend is almost the same. Conclusively, after alleviating the endogeneity, the differences between the returns to education in urban and rural China and their evolution have not changed significantly, indicating that the above-mentioned empirical conclusions have good robustness.

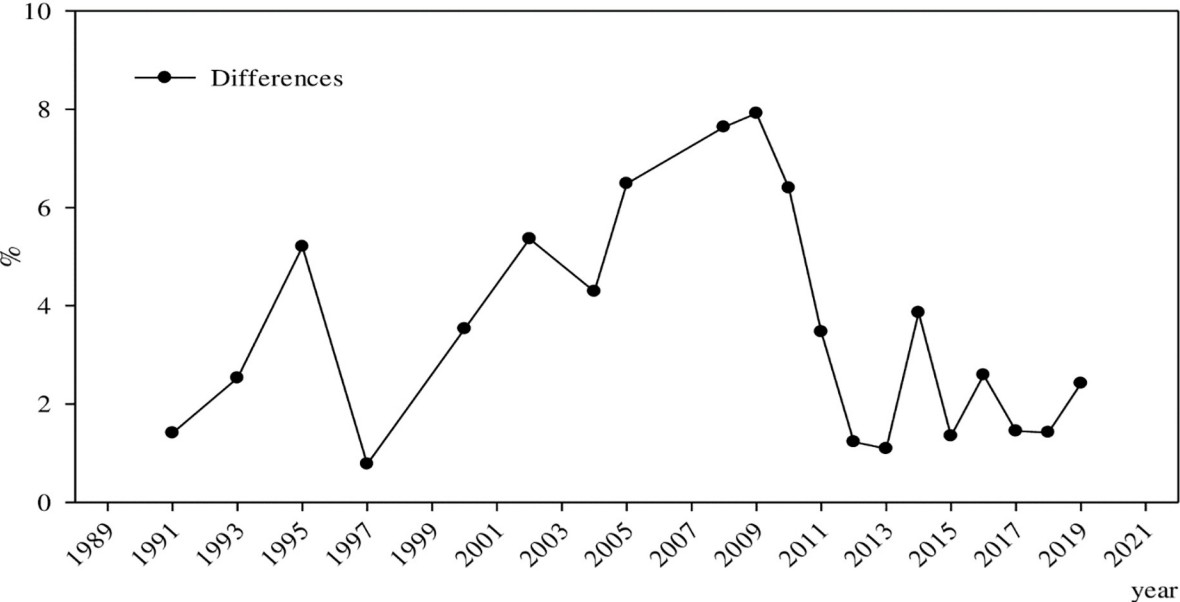

**Fig 4. Robustness test: Evolution of the differences in returns to education between urban and rural China.** *Data Source*: CHNS, CHIP, CGSS, CFPS, CHFS, and CSS.

## Reasons for the evolution of returns to education

### Marketization process of labor forces

The theoretical logic of the relationship between returns to education and labor marketization comes from Nee's theory of market transition. Nee (1989) pointed out that as the economy transitions into a market-oriented one, economic activities are gradually relying on the market mechanism for regulation, with weakened control from the government [35]. If the labor market is a market with free-flowing elements and perfect competition, the income obtained by an individual will be equal to their marginal productivity, and if an individual has received a good education, their labor productivity will be given full play. In reality, however, the labor market may not be perfectly competitive. According to Rosenzweig (1995), investment in education is not a "universal panacea" [36]. To obtain a return from such investment, we should rely on technological improvement or the reform of the market and political system. Even if the labor force is highly educated and has high labor productivity, it will be difficult for labor income to fully reflect human capital if the degree of marketization is relatively low and the efficiency of labor resource allocation is not high. Thus, if the effect of human capital is inhibited, the educational level of an individual will be higher, the income level will be improved, and their returns to education will be relatively lower. Zhao and He unveiled that the higher the marketization, the higher the return to education [37].

Before the reform and opening up, the government monopolized the labor market. In a centrally planned economy, where salary differences are determined by qualification or experience rather than productivity, the influence of educational level on salary differences is severely limited. In the 1984 *Resolution on the Economic System Reform*, enterprises were allowed, for the first time, to base their total salaries on their economic performance. After the State Council published *the Interim Provisions on the Use of Labor Contracts in State-owned Enterprises*, compared with permanent employment, the number of workers under labor contracts continued to increase [38]. However, given the absence of an effective labor market and social

insurance system, the scale and impact of these reforms are limited. In 1993, China set the goal of developing a socialist market economy. Since then, China's market economy has continued to expand. The promulgation of *the Labor Law* in 1995 also heralded improvements in the labor market, social security, and insurance reform. From the late 1990s to 2003, the state-owned sector experienced the most profound changes, with the privatization of small and medium-sized state-owned enterprises and the strict restructuring of large state-owned enterprises in strategic industries, including mass layoffs [39]. Concurrently, previously unknown non-state-owned and private enterprises have gradually become a powerful force in the economy and labor market. The change in the labor market has two significant effects on returns to education. First, non-state-owned enterprises can now easily set the salaries of workers in the fiercely competitive market environment, and education reflects higher productivity, which in turn leads to higher income. Second, non-state-owned enterprises increase the salary premium for skills in the non-state-owned sector by competing for talents and skills with the state-owned sector. Then, we explain the evolution of returns to education in urban areas in China by observing the changes in the level of marketization of labor forces in urban areas in China.

To analyze the changes in the degree of marketization of labor forces, how the marketization of labor forces is measured must be understood. Due to the differences in the indicators, measurement standards, calculation methods of indicators, and other aspects in the measurement of the marketization level of labor forces, the marketization indicators measured by different scholars also differ greatly. In some studies, only one or two indicators were used to measure the degree of marketization, such as the proportion of state-owned enterprises, non-state-owned enterprises in the total industrial output value [40], and the proportion of the number of employees in the state-owned sector or private sector in the total employment [41]. However, measuring with a single indicator only reflects a specific aspect of the market-oriented reform; comprehensively measuring the process of marketization is difficult. Hence, some scholars have created a comprehensive index of the degree of marketization on the basis of different factors, among which the marketization index was measured by Wang et al. and Li et al. The latest year of the marketization index data measured by Li et al. is 2008, whereas the latest year of the marketization index data measured by Wang et al. is 2017. The two marketization indexes are not directly comparable, but they do provide insight into the labor market process over time. Therefore, the characteristics of the changes in the degree of labor marketization in China can be roughly inferred by combining the two marketization indexes. Table 3 lists the two marketization indexes.

The change trends of the two marketization indexes are shown in Fig 5. Based on the evolution of the labor marketization index measured by Li et al., before 2005, the labor marketization index in China generally shows a gradual upward trend. However, the upward trend has not continued since 2005, decreasing slightly in 2006, and has basically stagnated. However, the marketization index measured by Li et al. is only up to 2008. Based on the evolution of the labor marketization index measured by Wang et al., the marketization index in China shows a slow growth trend between 2004 and 2007 and decreased significantly in 2008, perhaps due to the large-scale fiscal and monetary stimulus measures taken to reduce the impact of the 2018 global financial crisis, which enhanced the dominance of the state-owned sector in the economy [42]. Since then, the marketization index has only slightly increased, almost always remaining unchanged.

The comprehensive examination of the evolution of these two labor marketization indexes revealed that the degree of marketization of labor forces in China gradually increased from the late 1980s to the early and mid-2000s. Correspondingly, returns to education in urban China have been increasing overall during this period, which may have influenced the large difference in returns to education. Since the mid-to-late 2000s, the degree of marketization of labor forces

**Table 3. Labor marketization index in China (existing results are listed).**

| Year | Li et al. (2010) | Year | Li et al. (2010) | Year | Wang et al. (2017) |
|---|---|---|---|---|---|
| 1989 | 19.83 | 2002 | 64.76 | 2004 | 6.10 |
| 1990 | 20.25 | 2003 | 67.07 | 2005 | 6.12 |
| 1991 | 21.10 | 2004 | 70.53 | 2006 | 6.55 |
| 1992 | 26.04 | 2005 | 76.03 | 2007 | 6.92 |
| 1993 | 34.11 | 2006 | 75.19 | 2008 | 5.48 |
| 1994 | 37.72 | 2007 | 76.19 | 2009 | 5.53 |
| 1995 | 40.60 | 2008 | 76.40 | 2010 | 5.45 |
| 1996 | 41.43 | | | 2011 | 5.59 |
| 1997 | 49.93 | | | 2012 | 5.98 |
| 1998 | 55.49 | | | 2013 | 6.16 |
| 1999 | 55.29 | | | 2014 | 6.56 |
| 2000 | 60.64 | | | 2015 | 6.48 |
| 2001 | 64.26 | | | 2016 | 6.64 |

***Data Source***: *2010 China Market Economy Development Report*, Li et al. (2010); *Report on Market Index by Province in China*, Wang et al. (2017).

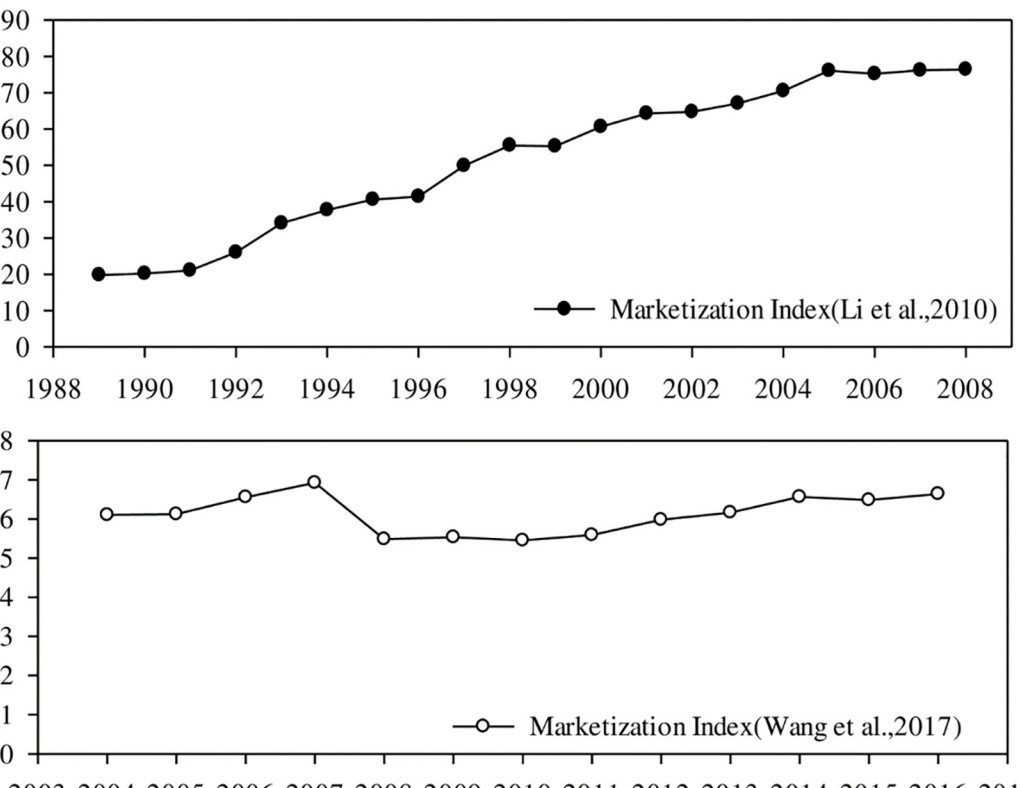

**Fig 5. Evolution of the degree of labor marketization in China.** ***Data Source***: *2010 China Market Economy Development Report*, Li et al. (2010); *Report on Market Index by Province in China*, Wang et al. (2017).

has not continued the previous trend. Even if no obvious downward trend exists, the growth has basically stagnated. The marketization of labor forces can no longer be the driving force behind the increase in returns to education in urban China, and as a result, the gap between returns to education in urban and rural China no longer has the potential to grow significantly larger.

## Changes in the relative supply and demand of labor forces

Changes in labor supply and demand will affect the income levels of different labor forces, thereby affecting the level of return to education. Based on the "race between education and technology" theory proposed by Goldin and Katz, when the relative demand for high-skilled labor forces grows beyond the relative supply, returns to education will increase [43]. Contrarily, when the relative supply of high-skilled labor forces grows beyond the relative demand, returns to education will decrease. Furthermore, Goldin and Katz discovered that returns to education in the United States fell in the 1970s before rapidly rising in the 1980s, owing to a large number of labor forces with high educational levels entering the labor market in the 1970s, exceeding the relative demand for labor forces with high educational levels. In the 1980s, the relative demand for labor forces with high levels of education in the labor market continued to rise, whereas the relative supply of labor forces with high levels of education remained essentially unchanged. Choi and Jeong's study on South Korea and Kambayashi et al.'s study on Japan unveiled that changes in the return to education are correlated with the supply and demand of workers with varying levels of education [44, 45].

Higher education in China has grown rapidly since the university entrance examination was reinstated in 1977. In 1977, the number of students enrolled in institutions of higher education was only 270,000, which increased year by year to 1.08 million in 1998 [46]. However, the rate of enrollment in institutions of higher education is relatively low during this period. Since the 1980s, two major events have transpired in China's education system, namely, the reform of compulsory education and the expansion of higher education. The first educational event occurred in the mid-1980s, when the government implemented the nine-year compulsory education. The enrollment rate of elementary schools reached 97.8% in 1990. The enrollment rate of junior high schools, on the other hand, developed slowly in the late 1980s but recovered in the 1990s. By 2000, the enrollment rate of junior high schools had reached 88.6%, and there were 16 million junior high school graduates, 5 million more than that in 1990. The second educational event occurred in the late 1990s, when the government implemented a policy of increasing education spending, which greatly expanded the scale of higher education, to mitigate the impact of the 1997 East Asian financial crisis. Between 1999 and 2009, the rate of college and university enrollment in China increased by four times, which is equivalent to the enrollment rate in the last 22 years, from 1978 to 1999. The number of graduates of higher education institutions also increased from less than 1 million in 2000 to 5.8 million in 2010, increasing by nearly six times in 10 years. Based on the data published in *the National Statistical Bulletin on the Development of Education in 2019* by the Ministry of Education, the consolidation rate of nine-year compulsory education in China reached 94.8%, and the gross enrollment rate of higher education reached 51.6%. The popularization of compulsory education has laid the foundation for the accumulation of human capital for the subsequent high-speed economic development in China. Since the mid-to-late 1980s, the supply of labor forces with secondary educational levels has continued to increase. Given the increase in the enrollment rate of higher education in 1999, numerous labor forces with high educational levels will enter the labor market, resulting in a substantial increase in the supply of labor forces with high educational levels. The increase in the relative supply of labor forces is not conducive to the improvement of returns to education.

However, technological progress has played an increasingly important role in economic growth, and it should be supported by high technology and high-quality talents to reduce the adverse effects of enrollment expansion to a great extent. After China's accession to the World Trade Organization in 2001, exports increased significantly. Meanwhile, a significant amount of foreign capital has also been invested in China. The technology and management levels of those foreign trade enterprises facing the international market were higher than those in the domestic market, whereas the technology and management levels of most foreign-funded enterprises were higher than those of local enterprises in China at that time. As a result of the increased number of foreign trade enterprises and the entry of a large number of foreign-funded enterprises, more highly educated and high-quality labor forces are required. To attract competent and highly skilled employees, the salaries paid by foreign-funded enterprises were much higher than those by domestic enterprises [47]. Since the end of the 20th century and the beginning of the 21st century, China's demand for highly educated and high-quality labor forces has expanded in tandem with the supply of such labor forces. The increase in the relative demand for labor forces can promote an increase in returns to education. Ding et al. pointed out that although directly observing the demand for labor forces with different education levels is difficult, it can be inferred from the increase of labor forces over time in different economic industries [48]. They found that 21.6 million job opportunities were created between 2003 and 2011 in the secondary industry, including mining, manufacturing, construction, and public utilities, accounting for 63% of all the job opportunities created. In 2003, the manufacturing industry had already become the economic sector with the most job opportunities, and it remained first throughout the period. By 2011, construction had become the second-largest sector. Contrastingly, the performance of the profitable tertiary industry with high added values for highly skilled labor forces was generally poor, with less than 8 million new workers over the same period. The rapid economic growth in China has not translated into a massive demand for labor forces with relatively high skills; hence total employment in these industries has remained stable or even fallen. Contrastingly, economic growth has been driven primarily by the expansion of sectors with low added values, demanding more unskilled or semi-skilled labor forces. Yang and Ding measured the labor demand with the proportion of the increase of the production value in the tertiary industry to the increase in GDP [49]. The ratio of the proportion of the employment with a college education and above to the proportion of the employment with senior high school education and the ratio of the proportion of the employment with senior high school education to the proportion of the employment with junior high school and below represent the relative supply of higher education and the relative proportion of senior high school education, respectively. In the early 2000s, the relative demand for labor forces in the urban labor market in China exceeded the relative supply of labor forces. Since the mid-2000s, the relative supply of labor forces with higher educational levels increased, but the growth rate of relative demand for labor forces was basically consistent or increasing only slightly.

Conclusively, since the mid-1980s, the relative supply of labor forces in China has increased, the relative demand for labor forces at all levels in the labor market has also increased, and its increase rate is higher than that of the relative supply, which is conducive to the increase in returns to education in urban and rural areas. However, due to the existence of the urban-rural dual economic structure, the increase in returns to education in rural China is lower than that of urban China, leading to a widening difference between returns to education in urban and rural areas. This persisted until around the mid-2000s, when the relative supply of labor forces with high educational levels increased substantially as students from the increased university enrollment gradually entered the labor market after graduation, whereas the relative demand of the labor market for these labor forces with high educational levels did not increase accordingly, dampening the rise in returns to education.

## Reform of the household registration system

Returns to education in rural China are low mainly due to the institutional segmentation caused by the unique household registration system in China, which results in the low efficiency of labor distribution and a non-competitive labor market with low mobility [50]. In addition, rural residents are mainly engaged in activities dominated by traditional agricultural production techniques [51]. However, due to the household registration system, low-educated labor forces from rural areas who are employed in urban areas without household registration cannot equally enjoy public services, such as medical care and education for children [52, 53]. As a result, the cost of living in urban areas for these low-educated labor forces from rural areas will be extremely high. Labor forces from rural areas are discriminated against in terms of salary and career choice in the urban labor market [54, 55]. These will also adversely affect the improvement of returns to education. Every reform of the household registration system can help break down these barriers and is critical for rural residents to gradually integrate into urban life [56]. Zhao pointed out that investment in higher education is beneficial for rural residents for two reasons. First, higher education increases an individual's human capital, which leads to an increase in income. Second, it increases the likelihood of an individual with rural household registration obtaining urban household registration, thereby increasing welfare [57].

The higher the education level of rural residents, the higher their chances of obtaining non-agricultural employment [58]. With the reform of the household registration system, the urban-rural dual economic system has been weakened. Such weakening can change the situation so that rural residents can only engage in agricultural activities and allow them more opportunities to participate in non-agricultural jobs. Simultaneously, the cost of living for rural labor forces will decrease as they gain access to some public services in the urban labor market. When rural residents engage in non-agricultural jobs, their non-agricultural income will increase coupled with the decrease in living costs, thereby increasing their returns to education [59]. The increases in returns to education in rural areas will narrow the gap between the returns to education in urban and rural areas. On the one hand, although the urban-rural dual economic system has weakened, a substantial difference remains in the cost of living and job opportunities between labor forces in urban and rural China due to the household registration system [60]. On the other hand, although there may be a few small factories or some industrial and service industries in the rural industrial structure in rural areas, the number remains extremely small compared with that in urban areas. Agriculture, tourism, and natural resource industries remain the main industrial structures in rural areas [61], which may be one of the reasons for the slow rise in returns to education in rural China.

## Changes in the quality of education

Even if individuals receive the same level of education, differences in educational quality will exist, resulting in differences in human capital accumulation, eventually leading to differences in educational returns [62, 63]. Teachers and teaching conditions in urban education are far superior to those in rural education, and there will be a significant difference in productivity between urban and rural laborers with the same years of education. Therefore, the sustainability of the quality of education is a key factor affecting returns to education [8]. As can be seen from the development of rural education quality, as the state places a high value on rural education and continues to increase financial expenditure on rural education, today's conditions for running rural schools in China have undergone significant changes, with hardware equipment, living conditions, and other aspects of the schools significantly improved [64]. According to the *Basic Conditions of the National Education Development in 2018*, the conditions for running schools for compulsory education in rural areas, especially in poverty-stricken rural

areas, have been significantly improved. Currently, 309,600 compulsory education schools in China meet the 20 minimum requirements for school operation, accounting for 99.76% of the total number of compulsory education schools. Furthermore, the capability of teachers in rural education has been continuously improved through state-led initiatives, such as focusing on cultivation, expanding the scale, raising wages, and encouraging exchanges. More and more university graduates choose to teach in rural areas, which can reflect the improvement of education in rural areas. As the quality of rural education improves, returns to education in rural areas will increase, which in turn will help narrow the difference between returns to education in urban and rural areas. Although the quality of basic education in rural areas is improving and the proportion of rural students entering high-quality universities has also increased, a large gap remains between the levels of basic education in urban and rural areas. For example, the overall quality of rural preschool teachers is low, and they are not educated professionals. Many of them are in charge of teaching multiple subjects, and some of their knowledge are outdated. Furthermore, regional imbalances exist in teacher allocation, with their rights and welfare not guaranteed [65]. Compared with urban students, the proportion of rural students entering high-quality universities remains relatively low, and the rate of increase is slow, which makes it difficult to rapidly increase returns to education in rural areas [66].

## Conclusion and discussion

Combining data from CHNS, CHIP, CGSS, CFPS, CHFS, and CSS, this paper conducts an empirical analysis of the difference between returns to education in urban and rural China and its long-term evolution from 1989 to 2019. The results are as follows. First, returns to education in urban China have been consistently higher than that in rural China, which is in line with the conclusions of most previous studies. Second, returns to education in urban China first increase slowly, then rapidly rise before turning into a slow decline and gradually leveling off, whereas returns to education in rural China increase slowly before gradually leveling off. Finally, the difference between returns to education in urban and rural China first grows larger, then smaller, before gradually leveling off. We believe that the labor marketization process, relative supply and demand of labor forces, reform of the household registration system, and changes in the quality of education are important for explaining the changes in returns to education. In addition, we considered the spouse's education as the instrumental variable of an individual's education and used the IV method for the robustness test. The findings indicate that the empirical conclusions of this paper are robust.

Overall, we believe that we should pay more attention to the role of human capital, including education in the allocation of resources; increase all types of investment in education; improve the labor forces; accelerate the comprehensiveness of the household registration system reform; weaken the segmentation of the dual labor market in urban and rural China; eliminate discrimination in public services; lower the cost of investment in education for rural residents; and encourage rural residents to invest more in education to increase their stock of human capital, thereby preventing the differences between rates of return in education in urban and rural areas from growing larger in the future. In addition, we should improve the construction of China's labor market system, raise the level of labor force marketization, and fully exploit the value of human capital, such as education, in the labor market.

The possible marginal contributions of this paper are as follows: (1) Returns to education in urban and rural areas in China and their differences are statically and dynamically analyzed to enrich the research on the evolution of returns to education in rural areas. (2) In the existing research on dynamic trends, only a single micro database is used. However, a single database often covers only a small time span. In some studies, they even only compared returns to

education at two time points, which led to the limitation of the results of dynamic analysis. Thus, we attempted to combine data from several large domestic microsurveys in China to ensure that dynamic analysis reaches longer time spans and more consistency.

However, this paper has the following limitations: In the reason analysis for the evolution of returns to education, we only performed qualitative analysis rather than quantitative analysis. Therefore, finding suitable data and using appropriate empirical analysis methods to testify the reasons for the evolution of returns to education are the future research directions.

## Supporting information

**S1 Appendix.**
(DOCX)

## Author Contributions

**Conceptualization:** Maishou Li.

**Data curation:** Xing Gao.

**Formal analysis:** Maishou Li.

**Methodology:** Xing Gao.

**Resources:** Maishou Li.

**Software:** Xing Gao.

**Supervision:** Maishou Li.

**Writing – original draft:** Xing Gao.

**Writing – review & editing:** Maishou Li.

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
