## [Decision Letter · Decision Letter 0]

21 Apr 2022

PONE-D-22-03043The Difference Between Returns to Education in Urban and Rural China and Its Dynamic Changes From 1989 to 2019PLOS ONE

Dear Dr. Li,

Thank you for submitting your manuscript to PLOS ONE. After careful consideration, we feel that it has merit but does not fully meet PLOS ONE’s publication criteria as it currently stands. Therefore, we invite you to submit a revised version of the manuscript that addresses the points raised during the review process.

The referees and I see value in this paper, given the novelty that it brings to the literature on the differences in returns to education in urban and rural in China, which, and I do agree with the referees on this point, is a very interesting and important topic.  Given the clear expertise of the referees, I defer to their comments and will ask you to simply respond to them. However, my view is that the main areas of concerns, which you would do well to address in rewriting the paper, are as follows:

(1) make the introduction more compelling, focusing on the relevance of the differences in returns to education in China (R1).

(2) Please consider reorganizing your literature review, trying to be more critical and thus more informative (R1)

(3) describe in detail the data sources you use, their representativeness, and justify all your choices (i.e. why you use some waves and not other). Please provide the reader with detailed information also about how you merged the data, in order to increase the replicability of your results.  You can also devote one or more appendixes to this very important issue, if you prefer. (R1 and R2)

(4) I agree with both reviewers that the data and methodology sections require some rewriting. I suggest considering inflation in the definition of your dependent variable and provide a discussion of identification threats your model faces (endogeneity), and how you solve them (with an IV approach), introducing a new econometric model directly in section 2.1. (R1 and R2)

(5) I agree with Reviewer #1 that your empirical strategy (i.e., IV) does not allow to take into account properly potential endogeneity, as the instruments you use may have a direct effect on the dependent variable and hence the results cannot be interpreted as causal relationships. For more details, please refer to the comments of the Reviewers reported below. R1 suggests two alternative instruments that are worth exploring.

(6) If it possible, I encourage you to provide some empirical support, even suggestive,  to the mechanisms you describe. (R1)

Beyond the above highlighted points, as I noted previously the referee reports are all of high quality, so please make sure to respond directly to all the comments.

Congratulations on the work so far; I look forward to reading the revision.

We look forward to receiving your revised manuscript.

Kind regards,

Simona Lorena Comi

Academic Editor

PLOS ONE

Journal Requirements:

Reviewers' comments:

Reviewer's Responses to Questions

**Comments to the Author**

1. Is the manuscript technically sound, and do the data support the conclusions?

Reviewer #1: Partly

Reviewer #2: Yes

2. Has the statistical analysis been performed appropriately and rigorously? 

Reviewer #1: Yes

Reviewer #2: Yes

3. Have the authors made all data underlying the findings in their manuscript fully available?

Reviewer #1: Yes

Reviewer #2: Yes

4. Is the manuscript presented in an intelligible fashion and written in standard English?

Reviewer #1: Yes

Reviewer #2: Yes

5. Review Comments to the Author

Reviewer #1: The authors estimated the differences in returns to education in urban and rural China over the past 30 years and analyzed the underlying causes. This is a very important and interesting topic relevant to many contexts. However, I do have some comments, which hopefully will be helpful in improving the paper.

1. Introduction: Returns to education is an important topic, but the introduction is not compelling enough to highlight its importance and why it matters even more in China.

2. Literature review: First of all, my understanding is that literature review should be a critical review rather than a list of citations stacking on top of each other. In the meantime, I would suggest that the authors cite some recent studies on the returns to education. For example, the one by Liu and her coauthors titled “The returns to education in rural China: Some new estimates.”

3. Data: I would also suggest the authors to provide more detailed introductions about the data used in the study, especially when the authors declare their biggest contribution is to integrate various micro databases. BTW, what do CHNS, CGSS, CULS, and CHIP stand for? Are they all nationally representative datasets? Are they panel or pooled OLS? How did you combine them together? Please note the data sources for the tables and figures.

4. Measurement:

a) Did you take price into consideration when measuring income? I think it is necessary to do so given such a long time span studied.

b) BTW, how did you measure marketization in your empirical analysis?

c) Table 3: Please clarify whether they measure the same thing? Different indexes constructed by different scholars, unless using the same method and data, otherwise might be apples and oranges.

5. Methodology:

a) The authors should clarify the rationale for dividing the urban and rural subsamples, whether by hukou or by geographic location. Since China has a large number of migrant rural workers living in urban areas, if the division is by geographic area, the increased returns to education of those population are also attributed to urban returns to education, which may result in an underestimation of rural returns to education. Besides, the authors should test whether the differences of returns to education between urban and rural are significant.

b) IV: Spouse’s education level does not necessarily satisfy the exogeneity of the instrumental variable. The authors need to further justify why some already widely used instrumental variables, such as the Compulsory Education Law (Fang et al., 2012) or the Prohibition of Using Child Labor (Xie & Mo, 2014), are not adopted. In addition, appropriate models for the IV estimation need to be specified.

c) I was wondering what the results look like if you control for the marital status of individuals and their employment sectors.

d) How do your OLS and IV results compare with each other?

6. Mechanism: This paper explores the mechanism relatively well. However, it is essentially narrative hypotheses/arguments. Please do provide empirical evidence, whenever data available for quantitative analyses, to test your hypotheses, and thus to support your arguments.

7. Presentation:

a) Please transpose Table 1 so that it looks more compact and neat.

b) Table 2, Please report all results or at least provide results for other variables in appendix.

c) Figure 1: Please also report the confidence interval.

8. The writing, including the explanations of data, variable definition, models, needs to be improved. In addition, the vertical axes in the figures all lack units.

9. Others: What do you mean by inflection point?

Reviewer #2: In this paper, the authors used multiple major sources of survey datasets in mainland China to analyze the long-term dynamic changes in returns to education in urban and rural China as well as the changing urban-rural gaps in returns to education over a long span of time, 1989-2019. Overall, I found this paper well organized, clearly written, and has a potential contribution to the existing field of research. However, I have several major concerns and minor suggestions before this paper should be considered for publication.

My first primary concern is related to the combination of the multiple datasets from different institutions using different methods of data collection. In Section 2.2 Data sources, the authors claimed that they used China Health and Nutrition Survey (CHNS) for 1989, 1991, 1993, 1997, 2000, 2004, and 2009. As far as I know, up to now, CHNS has made 10 waves available, including 1989, 1991, 1993, 1997, 2000, 2004, 2006, 2009, 2011, and 2015. So, why the authors did not use the data from 2006, 2011, and 2015? Similarly, for the Chinese General Social Survey (CGSS), there are additional waves already coming out, CGSS 2015 and 2017. I understand that there are several overlapping years among the major social survey datasets, but please provide a full explanation and rationale of why choosing one is not the other. The starting point is to offer readers the full names and more details of each dataset so that for some of the dataset selections we would understand that they were based on the data availability. Moreover, most of the datasets, such as CGSS, are cross-sectional, but CFPS is a longitudinal dataset. Different survey datasets have different sampling strategies and weighting. More discussions are needed to see how authors made a balance among different datasets for consistent analysis. Also, in terms of the urban vs. rural, there is a potential concern as the study spans the past decades. As China has experienced substantial development of urbanization since the economic reform in the 1980s, what was considered rural in the 1990s may have become urban areas in the 2010s.

The second major point is to related to variables and analysis. In 2.3 Description of variables, it says that the dependent variable is annual income. First, based on the skewed distribution of annual income, the variable is supposed to be logged. Second, the variable of annual income needs to add CPI-adjusted to take inflation over years into consideration. As for the independent variables, age may need to add a square term as the relationship between age and income may not be linear. Moreover, the variable work experience is based on age minus years of education. When including education, age, and work experience, I suspect that there may be issues of multicollinearity. Please recheck the models. In addition, as the authors demonstrated in the discussions on how the hukou system plays a significant role, then why hukou is not a control variable in the models? Here, I recommend a relevant article: The influence of hukou and college education in China’s labour market, by Xiao and Bian (2017) published at Urban Studies.

Additional minor points and suggestions the authors may wish to consider is the elegancy of tables. For example, in Table 2, it is better to keep two decimals rather than four which will reduce the length of the numbers. For the column of difference, instead of presenting the exact numbers, it’s better to use asterisks to indicate whether they are significant (* p<.05, ** p<.01, *** p<.001).

6. PLOS authors have the option to publish the peer review history of their article (what does this mean?). If published, this will include your full peer review and any attached files.

Reviewer #1: No

Reviewer #2: No

---

## [Author Response · Author response to Decision Letter 0]

1 Jun 2022

Thanks to the valuable suggestions from Academic Editor and Reviewers on the manuscript. We have tried our best to revise the paper according to suggestions. Please refer to "response to reviewers" for detailed modification information.

---

## [Decision Letter · Decision Letter 1]

28 Jun 2022

PONE-D-22-03043R1The Difference Between Returns to Education in Urban and Rural China and Its Dynamic Changes From 1989 to 2019PLOS ONE

Dear Dr. Li,

Thank you for submitting your manuscript to PLOS ONE. After careful consideration, we feel that it has merit but does not fully meet PLOS ONE’s publication criteria as it currently stands. Therefore, we invite you to submit a revised version of the manuscript that addresses the points raised during the review process.

Upon my own reading, I found that this version of the paper still suffers from some limitations, mainly related to the explanation and discussion of the empirical strategy.

Here are my main comments to this version of your paper.

I believe that more discussion should be added to the methodology section, where you added the IV model. You should explain to the reader why an alternative approach is needed: why is *Edu* endogenous? What are the assumptions Z needs to satisfy in order to be a “good instrument”? Are they satisfied? Please, remember that IV is a Local Average Treatment Effect, who are the compliers in your samples? These points should be addressed in the methodological section. In this version of the paper, part of the discussion about the endogeneity is reported in the robustness section, please move the discussion in the methodology section.Rather than talking about “dynamic changes”, I suggest using “evolution/changes over time”, even in the title. To me, and I believe to applied economists in general, the term “dynamic” refers to the study of how today level of a variable depends on the level it had yesterday. Empirically this is assessed using dynamic econometrics models, which often include the lag of the dependent variable. You rather use repeated cross-sections and thus do not model the dynamic in returns to education econometrically.  Please add some confidence intervals to your Figures, they will help a reader in assessing the evolution over time of your coefficients.I do agree with reviewer 1 that you should support your reasoning in the “Reasons for the dynamic changes of returns to education” section with some indirect evidence or references to the literature. You do not need to produce original evidence, but you should ground your reasoning on other research, otherwise it appears to be speculation. To illustrate, a sentence like "The reform of the household registration system mainly has a greater influence on returns to education in rural China." is a strong one. Who say that this reform had a great influence? You? Somebody else? If so, you should quote the research. You go on saying: " Returns to education in rural China is low mainly due to the institutional segmentation caused by the unique household registration system in China, which results in the low efficiency of labor distribution and the non-competitive labor market with low mobility. " This is an even stronger affirmation: you are establishing a causal link from the household registration system to low returns to education. Is there evidence of the causal link? If so you should quote the papers/literature/research, otherwise, you should ton down your sentence and make is suggestive.  As you see, this section needs a substantial rewriting in order to be published, because, in its current form, it contains some affirmations that are not supported by direct or indirect evidence. As suggested also by reviewer 1, please explain in the paper the limitations of your paper.Finally, as suggested also by reviewer 2, the paper requires a thorough professional English editing to meet PLOS ONE publication criteria, since the language is not always clear: please seek independent editorial help before submitting a revision. Please submit your revised manuscript by Aug 12 2022 11:59PM. If you will need more time than this to complete your revisions, please reply to this message or contact the journal office at plosone@plos.org. Please include the following items when submitting your revised manuscript:A rebuttal letter that responds to each point raised by the academic editor and reviewer(s). You should upload this letter as a separate file labeled 'Response to Reviewers'.A marked-up copy of your manuscript that highlights changes made to the original version. You should upload this as a separate file labeled 'Revised Manuscript with Track Changes'.An unmarked version of your revised paper without tracked changes. You should upload this as a separate file labeled 'Manuscript'.If applicable, we recommend that you deposit your laboratory protocols in protocols.io to enhance the reproducibility of your results. Protocols.io assigns your protocol its own identifier (DOI) so that it can be cited independently in the future. For instructions see: https://journals.plos.org/plosone/s/submission-guidelines#loc-laboratory-protocols. Additionally, PLOS ONE offers an option for publishing peer-reviewed Lab Protocol articles, which describe protocols hosted on protocols.io. Read more information on sharing protocols at https://plos.org/protocols?utm_medium=editorial-email&utm_source=authorletters&utm_campaign=protocols.

We look forward to receiving your revised manuscript.

Kind regards,

Simona Lorena Comi

Academic Editor

PLOS ONE

Journal Requirements:

Reviewers' comments:

Reviewer's Responses to Questions

**Comments to the Author**

1. If the authors have adequately addressed your comments raised in a previous round of review and you feel that this manuscript is now acceptable for publication, you may indicate that here to bypass the “Comments to the Author” section, enter your conflict of interest statement in the “Confidential to Editor” section, and submit your "Accept" recommendation.

Reviewer #2: (No Response)

Reviewer #3: (No Response)

2. Is the manuscript technically sound, and do the data support the conclusions?

Reviewer #2: (No Response)

Reviewer #3: Partly

3. Has the statistical analysis been performed appropriately and rigorously? 

Reviewer #2: (No Response)

Reviewer #3: Yes

4. Have the authors made all data underlying the findings in their manuscript fully available?

Reviewer #2: (No Response)

Reviewer #3: Yes

5. Is the manuscript presented in an intelligible fashion and written in standard English?

Reviewer #2: (No Response)

Reviewer #3: Yes

6. Review Comments to the Author

Reviewer #2: It is great to see the revised manuscript has addressed most of the comments I raised in a previous round of review. But, there are additional suggestions for minor revisions:

1. The authors added a footnote of the five of the six dataset they used in the analysis, but it is essential to provide more than a simply web link to the dataset but more detailed introduction to the datasets. What are the full names of these datasets? Which institutions designed and collected the data? Are they cross-sectional or longitudinal? And why they are reliable data sources?

2. The authors added a line to indicate that CPI at provincial level has been considered, but it is necessary here to add a footnote of data source and a brief discussion how CPI was adjusted.

3. The authors claimed that “the division of samples is conducted based on respondent’s registered residence, that is, whether they reside in the urban area or the rural area”. This statement is confusing and misleading because where their hukou is registered can be different from where they actually reside in. Millions of workers from rural areas holding rural hukou are working as migrants in cities without their hukou registered in urban areas. If the authors intend to compare the Urban-Rural division based on geographical location, then it is better to divide the sample based on where the respondents reside in and include where their hukou registered (urban or rural) as a control variable.

4. For the section exploring reasons for the dynamic changes of returns to education, they are more like hypotheses with any empirical evidences to prove. I would suggest the authors either trying to find relevant literature to support the hypotheses or selecting a few years that data is available to testify the mechanisms you hypothesized.

5. Add a paragraph or two to address the limitations of this study and provide future research directions. For example, it may be impossible for you to testify all your hypotheses on the mechanisms, then leave them to future researchers to explore.

Reviewer #3: The authors have well addressed the reviewers' comments and made corresponding revisions, and the current version of the manuscript has basically met the standard for publication. To make some further improvements, there are a few suggestions. First, the manuscirpt needs language editing to improve the language and overall content. Second, the conlusion section could be further improved by adding content on the contributions and limitations of the study.

7. PLOS authors have the option to publish the peer review history of their article (what does this mean?). If published, this will include your full peer review and any attached files.

Reviewer #2: No

Reviewer #3: No

---

## [Author Response · Author response to Decision Letter 1]

5 Aug 2022

Please see the file "Response to Reviewers".

---

## [Decision Letter · Decision Letter 2]

30 Aug 2022

Differences Between Returns to Education in Urban and Rural China and Its Evolution From 1989 to 2019

PONE-D-22-03043R2

Dear Dr. Li,

We’re pleased to inform you that your manuscript has been judged scientifically suitable for publication and will be formally accepted for publication once it meets all outstanding technical requirements.

The reviewer raised a couple of lasts very minor comments. At this point, I believe the paper is publishable after these last suggestions have been addressed, thus I encourage you to address them before submitting the final version of your manuscript.

Kind regards,

Simona Lorena Comi

Academic Editor

PLOS ONE

Additional Editor Comments (optional):

Reviewers' comments:

Reviewer's Responses to Questions

**Comments to the Author**

1. If the authors have adequately addressed your comments raised in a previous round of review and you feel that this manuscript is now acceptable for publication, you may indicate that here to bypass the “Comments to the Author” section, enter your conflict of interest statement in the “Confidential to Editor” section, and submit your "Accept" recommendation.

Reviewer #2: All comments have been addressed

2. Is the manuscript technically sound, and do the data support the conclusions?

Reviewer #2: Yes

3. Has the statistical analysis been performed appropriately and rigorously? 

Reviewer #2: Yes

4. Have the authors made all data underlying the findings in their manuscript fully available?

Reviewer #2: Yes

5. Is the manuscript presented in an intelligible fashion and written in standard English?

Reviewer #2: Yes

6. Review Comments to the Author

Reviewer #2: This is a better version of the paper and produces some useful findings. I have a few additional suggestions.

1. It should be "Appendices" rather than "Appendixes"

2. In the section "Reform of the household registration system", I hope the authors would give some solid examples of the reform that contributed to the changes in returns to education. Also, it is necessary to clarify different concepts of the urban-rural divide. One concept is based on the region/sample: cities vs. villages. The other concept is urban household registration vs, rural household registration. The authors mentioned that higher education increases the likelihood of a rural resident with a rural type of household registration obtaining the urban type of household registration. In this sense, this rural resident will become a new urban resident. Then in the urban sample, the population would be more dynamic, including residents who were born with urban household registration, new urban residents who used to be rural residents but obtain urban household registration and rural residents who migrate to cities but remain rural household registration.

7. PLOS authors have the option to publish the peer review history of their article (what does this mean?). If published, this will include your full peer review and any attached files.

Reviewer #2: No

---

## [Editor Report · Acceptance letter]

1 Sep 2022

PONE-D-22-03043R2 

Differences Between Returns to Education in Urban and Rural China and Its Evolution From 1989 to 2019 

Dear Dr. Li:

I'm pleased to inform you that your manuscript has been deemed suitable for publication in PLOS ONE. Congratulations! Your manuscript is now with our production department. 

Kind regards, 

on behalf of

Professor Simona Lorena Comi 

Academic Editor

PLOS ONE